# Challenges and the Way Forward in Demand-Forecasting Practices within the Ethiopian Public Pharmaceutical Supply Chain

**DOI:** 10.3390/pharmacy12030086

**Published:** 2024-05-31

**Authors:** Arebu Issa Bilal, Umit Sezer Bititci, Teferi Gedif Fenta

**Affiliations:** 1Department of Pharmaceutics and Social Pharmacy, School of Pharmacy, College of Health Sciences, Addis Ababa University, Addis Ababa P.O. Box 9086, Ethiopia; teferi.gedif@aau.edu.et; 2Edinburgh Business School, Heriot Watt University, Edinburgh EH14 4AS, UK; u.s.bititci@hw.ac.uk

**Keywords:** challenges, data quality, finance, forecasting, workforce

## Abstract

This study delves into the challenges of pharmaceutical forecasting within the Ethiopian public pharmaceutical supply chain, which is vital for ensuring medicine availability and optimizing healthcare delivery. t It aims to identify and analyze key hindrances to pharmaceutical forecasting in Ethiopia, employing qualitative analysis through semi-structured interviews with stakeholders. Thematic analysis using NVIVO 14 software reveals challenges including finance-related constraints, workforce shortages, and data quality issues. Financial challenges arise from funding uncertainties, causing delayed procurement and stockouts. Workforce shortages hinder accurate forecasting, while data quality issues result from incomplete and untimely reporting. Recommendations include prioritizing healthcare financing, investing in workforce development, and improving data quality through technological advancements and enhanced coordination among stakeholders.

## 1. Introduction

Forecasting is about predicting future events based on a foreknowledge acquired through a systematic process or intuition [1,2]. Forecasting is important to ensure an uninterrupted supply of essential medicines. Demand forecasting in a health supply chain is defined as the continuing process of projecting which health commodity will be purchased, where, when, by whom, and in what quantities [3]. Demand forecasting is a well-developed scientific discipline but has not been fully used in improving health supply chains in Africa. As a result, supply chains that serve patients in Africa remain weak and ineffective, putting treatment programs at risk, and weakening the overall health system’s ability to respond to the healthcare needs of the population [4,5,6,7,8]. Even though there has been significant support for addressing global health issues such as HIV/AIDS, malaria, and TB in developing countries during the last decade, the availability of medicines remains very poor in public health facilities [9,10].

There are many problems in global health supply chains, including fragile last-mile delivery, human resource challenges, fewer opportunities for research and development, problems in procurement and forecasting, and the lack of an accountability structure [11,12,13]. These problems are more exacerbated in Africa due to the inefficient use of scarce resources, stockouts, corruption, product diversion, or poor last-mile delivery, leading to reduced health outcomes. These problems are exacerbated by the lack of accountability and fragmented leadership, making it easier to pass on the blame and create strong incentives for corruption [6]. Moreover, the continent also suffers from poor forecasting skills of health supply chain professionals, requiring better allocation of resources for capacity building and development [14].

Poor demand forecasting is a key driver of stockouts and raises the risk of substandard medicines entering the health system [15]. Forecasting challenges vary based on the nature and extent of risks faced by various stakeholders of health supply chains [16]. Some of these risks are emerging features such as change in disease burdens, complicated international markets, dynamic patient needs, new products, and advanced health technologies. The asymmetry of these risks can create missed opportunities for health supply chain stakeholders and hamper access to healthcare [17]. In addition, demand forecasting of healthcare commodities depends on exogenous factors such as the availability of data on disease burden or cultural and demographic influences. Moreover, weak budgets and budgetary constraints can impact contractual obligations, especially in a resource-constrained set-up. This vulnerability will impact demand forecasting, thereby creating gaps in the scientific and financial pipeline. The uncertainty of grant approval, complexity of disbursal cycles, and the sustainability of funding health commodities are unique risks embodying health supply chains. 

In the decision-making process, forecasts could be considered the cornerstones of effective strategies, since the data gathered from them scan either enhance or thwart the organization’s survival. There are multiple models of forecasting, and the various stakeholders must seriously consider which approach will provide the best information and the right guidance towards implementing that information successfully. Also, the quality of forecasting tools impacts the ability to gather adequate data about the future demands and trends [18]. The healthcare sector is continually evolving, which presents both opportunities and threats. It is difficult to standardize forecasting tools since health demands often differ due to factors such as patient experiences, resource allocation, disease burden, leadership. Appropriate forecasting tools will determine projections based on identified business drivers, influencing factors, and business constraints [19]. Moreover, the selection of forecasting methods depends on the availability of data, volatility of demand, and maturity of products [20]. Therefore, this study tries to identify what the challenges of forecasting practice are in the Ethiopian Pharmaceutical Supply Services (EPSS) and to provide strategies for mitigating the challenges related to forecasting practice. 

## 2. Literature Review

### 2.1. Challenges in Pharmaceutical Forecasting

One of the weakest links in global health supply chains is demand forecasting of health commodities. Many of the health supply chain bottlenecks impede accurate demand forecasting, and without the accurate demand forecasts, market viability, production capacity, and financial commitments become challenging. National governments and the international donor community rely on forecasts for budgeting while implementing partners and health programs depend on forecasts to plan their health supply chain logistics [18]. 

The challenges of demand forecasting in public health systems in Africa include limited funding, poor infrastructure, and lack of skilled personnel, which can hinder the implementation and maintenance of demand-forecasting systems [21]. Moreover, the challenges of unreliable data are also another issue; data collection for demand forecasting may be challenging in Africa due to issues such as incomplete or inaccurate data, lack of standardized data collection methods, and limited access to real-time information. Moreover, forecasting new health commodities is difficult due to uncertainties in how breakthrough healthcare technologies and products will be accepted by consumers [18]. There is a lack of information gathering to inform procurement and supply decisions. While health facilities see real demand daily, it is most often logged in a paper-based system and not shared with other levels of the supply chain [22].

The engagement and support of leadership at various levels of the health system are crucial for the successful implementation of demand-forecasting systems in Africa. A lack of accountability and fragmented leadership can create challenges in demand forecasting and supply chain management [6]. The current system of health delivery is siloed, fragmented, and uncoordinated; this lack of coordination will affect forecasting as well as the whole supply chain [22]. The lack of qualified personnel leads to high workloads and low performance while leaving key duties unattended. In fact, there are often insufficient trained supply chain staff at warehouses and health facilities to perform even basic duties [22,23]. High workloads, lack of training, deficient facilities, poor working conditions, and inadequate pay not only affect employees’ ability to perform their jobs but also affect morale and turnover [22,23]. Moreover, the quality of forecasting tools impacts the ability to gather adequate data about future demands and trends [18]. Shifting disease burdens, dynamic patient needs, and new products in complex international markets contribute to forecasting challenges [17]. Challenges such as stockouts, corruption, and product diversion can lead to reduced health outcomes and impact demand forecasting [7]. Resource-constrained settings face major logistic challenges in forecasting demand for health commodities [18]. Limited information on how risks are allocated and incentives are aligned among stakeholders can impact demand forecasting in health supply chains. Healthcare organizations often operate within budget constraints, which can impact their ability to accurately forecast and meet the demand for medical technologies [24]. The robustness of health forecasting tools has also been mentioned as one of the challenges in health forecasting’s [25]. 

The pharmaceutical industry’s demand forecasting often relies on simplistic statistical methods using historical data which fail to capture complexities influenced by economic conditions, regulatory changes, market competition, special contracts, and media attention. Addressing these complexities requires advanced models incorporating extensive data and industry expertise [26]. Demand sensing has emerged to consider these factors, highlighting the need for sophisticated forecasting approaches [27]. Simple time series models assume future demand depends only on past trends, neglecting these multifaceted influences. Maroun et al. categorized forecasting challenges as internal and external. Internal challenges arise within the organization and include employee, management, communication, and cultural issues, hindering information sharing and collaboration, thus affecting forecasting accuracy. External challenges such as market dynamics, customer behavior, and supplier relationships come from outside the organization, introducing uncertainties and market fluctuations. Internal challenges impact organizational processes, while external challenges are driven by market forces and environmental factors [28].

Traditional statistical models like exponential smoothing have become inadequate in dynamic and volatile post-2020 environments. Forecasting in such uncertain settings is challenging due to the complex and evolving data requirements, often termed as a “wicked problem” [29,30,31]. Unforeseen events like the COVID-19 pandemic can disrupt forecasting accuracy, resulting in shortages and excess inventory [32]. Despite advancements in analytics and Artificial Intelligence (AI), quantitative forecasting models may struggle to handle the complexity of changing data requirements. While machine learning methods are valuable, human interpretation, context awareness, experience, intuition, and creativity remain essential for detecting non-trivial patterns in forecasting [33,34]. This highlights “Moravec’s Paradox”, where human judgement and analytics possess unique strengths and weaknesses [35,36,37,38,39,40]. An ideal forecasting methodology integrates both judgmental and statistical approaches, leveraging collective intelligence to enhance accuracy [37,41,42]. Demand uncertainty, which is small variations in patient demand at health clinics, can lead to significant fluctuations in order quantities throughout the supply chain, resulting in stockouts or excess inventory. Ineffective and limited forecasting capabilities due to imprecise forecasts in procurement planning can result in stockouts or product expiration [6].

### 2.2. Enablers That Can Enhance Demand-Forecasting Practices in Health Supply Chains

Enabling factors for improved demand forecasting in health supply chains encompass access to accurate consumer data, reliable sources, and data quality assurance [43]. Public–private partnerships and collaboration enhance forecasting by pooling resources and expertise [44]. Investment in skilled personnel and transparent regulatory frameworks bolsters forecasting capabilities and reduces risks [45]. Information sharing among stakeholders and utilization of technologies such as data analytics enhance efficiency and accuracy [18]. Integrated IT systems streamline processes and mitigate errors in forecasting [28].

Involving key stakeholders improves forecasting outcomes, while continuous monitoring and evaluation enhance accuracy [43]. Standardized approaches ensure consistency in health forecasting, facilitating comprehensive implementation and optimal resource utilization [22]. However, achieving coordination poses challenges due to diverse actor priorities [6,46]. Enhanced communication within organizations fosters better coordination in demand forecasting [28].

Encouraging data-driven adjustments reduces bias and enhances forecast accuracy, with past adjustments informing future decisions to minimize biases. Considering external factors enhances demand predictions, ultimately enhancing organizations’ ability to forecast uncertain demand effectively and overcome challenges [28]. This study aims to identify demand-forecasting challenges in Ethiopian public pharmaceutical supply chains and propose strategies for their alleviation, addressing the following research questions: (1) identifying prevailing challenges in demand forecasting; and (2) suggesting effective strategies to mitigate these challenges.

## 3. Methods

### 3.1. Research Design

This study utilizes a cross-sectional case study research design to assess the challenges of the pharmaceutical forecasting practice in the EPSS.

### 3.2. The Context

EPSS plays a pivotal role in providing quality pharmaceuticals to public health facilities across Ethiopia, serving a population of 105 million through 19 branch warehouses and over 3800 health facilities [47,48]. EPSS manages an annual turnover of nearly USD 1 billion, overseeing the procurement, warehousing, and distribution of pharmaceuticals and medical supplies [48].

Despite its significance, EPSS faces challenges such as inaccurate drug forecasting, and procurement delays compounded by data quality issues and communication gaps within EPSS [49]. A study identified human, financial, infrastructure, and technological challenges within the pharmaceutical supply chain [50]. EPSS struggles with systemic weaknesses, including inadequate demand planning, equipment management, and information systems [48].

A SWOT analysis highlighted concerns such as low customer satisfaction, weak supplier relationships, inventory management issues, and distribution planning deficiencies. Addressing these challenges necessitates efforts to enhance forecasting, procurement, data quality, and overall supply chain management. EPSS operates two pharmaceutical supply streams: the Revolving Drug Fund (RDF) and program pharmaceuticals. RDF quantification begins at health facilities, with requirements consolidated at EPSS branches and centrally. Program commodities are quantified centrally, with supply plans based on forecasting outputs [48].

### 3.3. Reflexivity

All authors of this study were not part of the EPSS. Three of team members (A.B., T.G., and U.S.) had extensive experience and expertise in qualitative research methodology. The interviews were conducted by one of the local authors (A.B.). The authors had previous training in and a good understanding of the study context and values; they performed the transcription and crosschecked the contents embedded in the translation. An initial analysis of potential informants was discussed by the authors, and schedules were created to carry out the interviews at the participants’ offices.

### 3.4. Participants

A maximum variation sampling method was considered to recruit a representative number of participants from all regions of the country. Health professionals and top management members involved in health supply chain services, forecasting, and decision-making at the hospital level as well as the regional level were purposely selected to ensure that appropriate informants would provide rich study data. A total of 17 interviews were conducted with professionals from different sectors which have a direct or indirect relation to pharmaceutical forecasting practices. Accordingly, nine in-depth interview participants were from regional health bureaus and worked either in leadership or in health services and had direct contact with regional forecasting activities. Health facilities located in potentially eligible regions were contacted. Participants were requested to consent to being included in this study. Four interview participants were from the EPSS, where they are directly working with forecasting units and perform forecasting-related activities. Further, we included two participants form partner organizations who had direct roles in forecasting activities or supported these activities. Two interview participants were from the Ministry of Health and had a direct role in coordinating as well as monitoring forecasting activities at the Ministry of Health of Ethiopia. To ensure the anonymity of study subjects during analysis, each participant’s identifier was either coded with labels or altered.

### 3.5. Data Collection Methods and Procedures

A semi-structured interview guide was developed and used for all the interviews. Following identification and consent, informants were approached on the scheduled dates to freely share their opinions. Interviewers had no direct connection with the interviewees, but were immersed in the situation so as to prevent interviewees from feeling evaluated, coerced, or that they were risking something by taking part in this study. At times, participants shared their experience over the course of the study with the researchers. They were also amicably informed that they would be part of this investigation as appropriate personnel in the healthcare setting. The phenomenon of interest in this study was the challenges of forecasting practice in the Ethiopian public supply chain system. The following questions were asked: “How do you see the health supply chain in Ethiopia and in particular in EPSS”, “What are the factors which affect the availability of pharmaceuticals in Ethiopia and in particular with EPSS?” Further questions and probes to help participants elaborate on their views and on the facets that shaped them were also raised. Among others, these included the following: “Explain the problems in forecasting and ask why there is huge forecasting problems in EPSS? Discuss on the solutions which helps to improve on the availability of pharmaceutical in Ethiopia and in EPSS (Probe what to be done regarding forecasting problems of pharmaceuticals in EPSS). And what role does data quality and availability play in forecasting accuracy, and how are data-related challenges addressed? Can you describe the collaborative efforts and partnerships among stakeholders involved in forecasting and supply chain management for pharmaceuticals in Ethiopia?” One-to-one interviews with key informants were conducted during the specified period and lasted 40–60 min. All interviews were digitally audio-recorded and transcribed verbatim. Data collection was conducted until a point of theoretical saturation. 

### 3.6. Ethical Approval

Ethical approval was obtained from the Institutional Review Board of Addis Ababa University, College of Health Sciences, and the Ethics Review Committee of the School of Pharmacy, Addis Ababa University (Protocol number: 010/22/SoP). Permission was also granted by EPSS Directors. Participants were assured of their right to withdraw, and written consent was obtained. Sessions were conducted in private, and pseudonyms were used to protect participant identities. Tape records and transcripts were securely stored and kept confidential.

### 3.7. Data Analysis

The transcribed data were cleaned and imported to NVIVO 14 Qualitative Data Analysis software. Analysis was conducted following the steps stipulated by Braun and Clarke (2006) [51]. First, we repeatedly read the textual transcriptions for familiarization; then, initial codes of relevant data were generated. Related codes were then merged into themes, which were reviewed for coherence and data support, which led to condensed themes. On this basis, we defined the final themes and produced a composite description of the essence. Themes were modified and refined inductively from the data in Microsoft Excel through a thematic content analysis approach guided by the descriptive social constructivism framework. The dynamic interactions across the professional, patient, stakeholder, institutional, and system levels that could influence providers’ views of the phenomenon were described. The validity of the findings was verified by employing various methods. These included undertaking a thorough literature search, following steps that ensure a trustworthy thematic analysis as suggested by Brien et al. (2014), bracketing of the researchers’ experience of the phenomenon from the participants’ view, and crosschecking of contents among the researchers [52].

## 4. Results

### Socio-Demographic Profile of Respondents

A total of seventeen participants were successfully interviewed, of which fourteen were male and three were female. Most of the participants were over 35 years of age, with more than 10 years of professional experience. More than 80% of the participants had a master’s degree in the field of health supply chains (Table 1).

**Table 1 pharmacy-12-00086-t001:** Socio-demographic profiles of respondents as of 2023.

Gender	Frequency	Percentages
Male	14	82.4
Female	3	17.6
Age of the respondents in years		
25–35	7	41
>35	10	59
Years of experience		
3 to 10 years	7	41
More than 10 years	10	59
Level of education		
Bachelor’s degree	3	17.6
Master’s degree	14	82.4

In this study, two main themes and ten sub-themes were developed based on content analysis. The first theme was challenges related to forecasting in the EPSS and the sub-themes were (1) finance-related, (2) workforce-related, (3) data quality-related, (4) technology-related, (5) coordination-, collaboration-, and leadership support-related challenges. The second theme were strategies for improving forecasting, and the sub-themes were (1) responsibilities of and attention paid to forecasting; (2) improvements in data quality; (3) workforce- and capacity-building-related challenges; (4) finance-related challenges; (5) stakeholder role-related challenges (Figure 1).

Theme 1: Challenges related to forecasting in the EPSSSub-theme 1: Finance-related factors

Participants emphasized widespread finance shortages for procuring pharmaceuticals, exacerbated by delayed budget releases post-quantification exercises. At the central EPSS level, pharmaceuticals are provided to health facilities on credit, exacerbating financial constraints. Participant 3 illustrated this as follows:

*“Health facilities are buying the product on credit based from EPSS and the payment period is lengthened and this has created a finance shortage in EPSS where EPSS cannot be able to open Letter of Credit at the bank due to the shortages of local currency and un collected sale from the health facilities”* (Participant 3). 

At the health facility level, delays in budget releases and uncertainty regarding allocated budgets for pharmaceutical procurement were common concerns. Participants noted that quantification exercises were often conducted without knowledge of the budget, leading to uncertainty in product purchases. For instance, Participant 8 illustrated this as follows:

*“Most of the time the facilities quantify their need without having adequate budget. So the health facilities they only send their demand but they are not sure they will purchase those specific products from EPSS. Due to this EPSS may also purchase either small or big amounts of the products”* (Participant 8).

Additionally, disparities between the timing of budget releases and quantification exercises further exacerbated finance-related challenges, as stated by Participant 11:

*“Budget shortage we don’t know how much is the budget moreover the time where we conduct the quantification and the time where the budget release is different. For instances for The budget release might goes until October 2024 however the quantification is done on April 2023 and these is a major challenge”* (Participant 11).

Furthermore, shortages of hard currency for pharmaceutical imports compounded financial challenges at the central EPSS level. Nearly all participants mentioned the difficulty in obtaining sufficient hard currency due to Ethiopia’s import dependency for pharmaceuticals. This issue was exacerbated by disparities between forecasted and available funds, as described by Participant 6:

*“Capacity of the country is huge and it is not possible to get sufficient hard currency for the pharmaceuticals fund. In programme medicine we forecast on central based we have may stake holders which is good the problem is the fund for instance we forecasted for maternal health medicines around 8 million dollar however the available fund for the product is 1.8 million dollar so at that time we reconcile the forecast. You can imagine the frustration it will create”* (Participant 6).

Finance shortages were evident for both RDF and program pharmaceuticals, impacting procurement despite adequate forecasting. Participant 2 emphasized that funding shortages, rather than forecasting problems, were the main reason for national-level commodity shortages, underscoring the critical importance of addressing funding availability issues.

*“Capacity of the country is huge and it is not possible to get sufficient hard currency for the pharmaceuticals fund”. “In the health programme in my evaluation there is no forecasting problems, I am doing in HIV, Malaria, family planning Maternal and Child, and TB programme. There is a lot of investment in this area, from different funding organizations, there are well capacitated staffs in the area and I can say there is no forecasting problem at all. But what we forecasted will not be purchased due to the challenge of funding so when there are shortages of funding we will prioritized and focus on the most impactful commodities due to the shortages of funding so we cannot say that it is due to the forecasting problems rather it is due to the funding availability problems.”* (Participant 2).

Similarly, Participant 16 noted finance shortages impeding RDF pharmaceutical procurement despite quantification efforts. Addressing these finance-related challenges is imperative to ensure consistent pharmaceutical availability within the Ethiopian public pharmaceutical supply chain.

*“Even if they quantify RDF pharmaceuticals we cannot be able to purchase due to the shortage of finance”* (Participant 16).

Sub-theme 2: Workforce-related factors

Participants highlighted workforce challenges impacting forecasting, including shortages and competency issues. The attrition of skilled professionals and limited institutional emphasis worsened the situation. The scarcity of proficient forecasters was evident both locally and nationally, reflecting systemic issues. Shortages of quantification experts and lack of specialized training posed significant obstacles. Additionally, delegation of forecasting to nurses led to inconsistent data quality. Participant 17 noted supply chain workforce shortages, while Participant 13 emphasized the prevalence of low-skilled human resources, mainly nurses.

*“There is a shortage of work force in the supply chain”* (Participant 17).

Another respondent said the following: 

*“There is low skilled human resources related with low capacity and most of them are nurses”* (Participant 13).

Participant 8 lamented the absence of pharmacy professionals equipped with the technical acumen necessary for accurate quantification. Additionally, Participant 2 identified inadequate manpower capacity as a primary impediment to timely and accurate forecasting within health facilities. 

*“There is no pharmacy professionals at the health facilities and have no the technical skills to quantification”* (Participant 8).

*“Man power capacity, to request and report what is needed on time at the health facility is one of the main challenges in forecasting’s”* (Participant 2).

Sub-theme 3: Data-related challenges 

Participants highlighted challenges with data quality in the Ethiopian pharmaceutical supply chain. Poor data from health facilities, compounded by limited data visibility across the supply chain, hinder effective forecasting. Participant 8 emphasized false reporting and incomplete data, revealing a knowledge gap among health professionals on reporting protocols, complicating accuracy.

*“There is false reporting which came from the health facility and there is a data quality problem. The data feature is not complete, as well as timely. Some of the health facility reports only when they are stock outs of the products and they have a knowledge gap as the IPLS indicated that there should be a forced reporting system to be applied even if you have the products you need to report that however most of the professionals do not comply with this methods”* (Participant 8).

Participant 2 raised concerns about inadequate information gathering and reporting practices, noting deficiencies in demand generation and commodity requesting procedures. Participant 4 also questioned the reliability of data from lower-tier health systems, impacting product availability. 

*“There is improper information gathering, improper demand generation and improper reporting and requesting of commodities at respected hubs”* (Participant 2).

*“Challenges related with the quality of data the data is not reliable especially those data which comes from the lower health systems and these have impact on the availability”* (Participant 4).

Moreover, Participant 13 raised concerns about outdated bin card records and the absence of formal recording systems in rural health facilities, noting the reliance on assumptions in lieu of accurate data. 

*“There is a problem of data, bin card are not updated, and there is no recording system in most of the rural health facilities and most of the things are based on assumptions and these will create a problem in forecasting’s”* (Participant 13).

Compensatory measures employed by experts at the EPSS were also scrutinized, with Participant 17 highlighting the tendency to over-quantify and prioritize forecasting for high-patient-load facilities. Such practices, while intended to mitigate data inaccuracies, risked distorting procurement priorities and exacerbating forecasting errors. 

*“There is a problem in the data quality for forecasting and there is over quantification beyond the budget capacity. The forecasting will not cover all the health facilities and only for the most high patient load only get focus from EPSS as well as from our side for the rest of the health facilities we do extrapolation and these action will affect the forecasting”* (Participant 17).

Participant 5 highlighted data reliability challenges, noting adjustments made due to perceived inaccuracies. Participant 3 stressed the importance of health professionals to data integrity, lamenting speculative reporting over evidence-based methods, undermining forecasting and supply chain management effectiveness.

*“The data which comes from the health facility is a false data so what we do is we add all and we divide it by two”* (Participant 5).

*“One of the challenges in forecasting is the emphasis that we gave for data is so poor and unable to understand the significance of data. For example EPSS collect data from the health facility and uses for forecasting if professionals working on the health facility did not understand the significance of the data and send halfhazardly since this data will have impact on the forecasting it will create a huge forecast inaccuracy. Most of the health professionals when they send their data they just guess how much they need it rather than focusing either historical data that they have. Since the data they send us is not correct that affected the whole process of the forecasting’s”* (Participant 3).

Sub-theme 4: Technology-related factors 

Participants highlighted challenges including inadequate technologies and a lack of data visibility affecting forecasting. At the central EPSS level, participants noted a lack of end-to-end visibility into pharmaceutical availability at health facilities, causing forecasting and availability issues. Additionally, participants mentioned using Excel-based forecasting tools, which were deemed user-unfriendly.

This was summarize as follows: 

*“Does not have a good tool for doing forecasting”* (Participant 14). 

Another respondent also described this as follows:

*“There were no standardized tools however currently we have standardized the tools at the national level”* (Participant 7).

Sub-theme 5: Coordination-, collaboration-, and leadership support-related factors 

Participants emphasized the importance of coordination, collaboration, and leadership in pharmaceutical forecasting, crucial for enhancing pharmaceutical availability. Leadership engagement, especially at the political level, emerged as a key factor influencing forecasting effectiveness. Participant 16 highlighted the impact of leadership support on pharmaceutical availability and quantification, noting that inadequate political backing could hinder effective forecasting. Similarly, Participant 17 lamented the lack of attention given to supply chain management compared to other health programs, reflecting broader political disparities.

*“Leadership level impact the on the availability as well as the quantification of pharmaceuticals, where quantification is not supported by the political leaders”* (Participant 16).

Other respondents described this as follows:

*“The political leaders do not understand and support the pharmaceutical supply chain for example maternal and child health has a good attention every time there is reporting of how many mothers gave birth, so the supply chain do not have such attention as the other programme”* (Participant 17).

Participants noted organizational coordination challenges in managing quantification processes. Shifting responsibilities between the Ministry of Health Pharmacy Unit and the EPSS Quantification and Market Shaping Unit led to coherence and coordination issues. Participant 16 highlighted mandate-related discrepancies, noting adverse effects on forecasting accuracy due to inconsistent representation within the EPSS. The absence of EPSS representatives during quantification meetings worsened coordination challenges. Concerns were raised about the efficacy of Drug and Therapeutics Committees (DTCs) in supporting quantification exercises, as their involvement was deemed inadequate. Pharmacy professionals bore the burden of quantification, leading to various issues. Strengthened coordination, enhanced leadership support, and improved collaboration among stakeholders are needed to address these challenges in pharmaceutical forecasting within the Ethiopian public health system. 

*“There are mandate related challenges some time it the quantification exercise was conducted by EPSS some other time it comes to PMED these make lack of coordination”. Those forecasting professionals who were to represent EPSS did not attend the recent quantification for the last two meetings you can imagine the impact of these”* (Participant 16). 

Theme 2: Strategies for improving forecasting

The participants delineated several strategies to address the challenges associated with pharmaceutical forecasting. These strategies can be categorized into distinct themes, with the overarching goal of enhancing the management of the quantification process. The identified sub-themes encompass (1) responsibilities of and attention paid to forecasting, (2) improvements in data quality, (3) workforce and capacity-building initiatives, (4) finance-related interventions, and (5) stakeholder engagement.

Sub-theme 1: Responsibilities of and attention paid to forecasting

Participants emphasized the lack of attention given to forecasting in supply chain management, stressing the need for earnest attention from management, DTCs, and political leadership. They highlighted the imperative for heightened awareness and support for forecasting within the healthcare sector. One participant articulated the need for equitable attention to the health supply chain, akin to other programmatic initiatives, advocating for proactive consultation during instances of product stockouts and advocating for sectoral prioritization. Their perspective is encapsulated in the following statement: “*There needs to be attention given to the health supply chain just like other programs. We even consult when there are stockouts of products, and due attention should be given to the sector*” (Participant 13). Echoing this sentiment, another respondent highlighted the importance of incorporating quantification as a Key Performance Indicator (KPI) and underscored that resolving forecasting challenges necessitates commitments at higher organizational echelons. This viewpoint is elucidated by Participant 4, who emphasized, “*Including quantification as a KPI is important, and the challenges of forecasting will not be resolved with professionals alone; it requires higher-level commitments*”.

Participants noted attention disparities among pharmaceutical categories like program and RDF (Revolving Drug Fund) drugs, highlighting stakeholder engagement variations. One participant emphasized unequal support for programmatic versus RDF pharmaceuticals supply chains, urging equal focus. They stressed product availability’s impact on program effectiveness, especially due to inadequate supply chain infrastructure. Participant 13 articulated this perspective, stating, “*For programs, there is high support; however, for the supply chain, there is no attention. Just like programs, there should be due attention for the supply chain, as without product, there is no program*”. Furthermore, they underscored the adverse ramifications of inadequate supply chain frameworks, particularly within non-governmental organizations (NGOs), which often lack dedicated supply chain units, thereby exacerbating product stockouts and impeding programmatic effectiveness.

Sub-theme 2: Data-related improvements 

Participants stressed the crucial need for data quality improvements, emphasizing accuracy, completeness, and timeliness. Robust data were deemed essential for informed decision-making and strategic planning. Participant 14 highlighted this, stating, “*Data quality is crucial for all decisions*”. Echoing this perspective, Participant 10 advocated for the establishment of end-to-end data visibility. “*There should be end to end data visibility where it will indicate the point of sale data when there is a sell of pharmaceuticals. The data should be visible at Centre, hub and facility level*”. Their assertion underscores the importance of leveraging technological advancements to ensure seamless data accessibility and integrity throughout the supply chain continuum. Similarly, Participant 15 called for improved data visibility and technological integration, expressing skepticism about data reliability from health facilities. They emphasized control mechanisms and innovative technologies for enhanced accuracy. “*We need to ensure end to end data visibility and introduce new technologies and I don’t trust the data which is coming from the health facilities and we don’t have any controlling system of the data coming from the health facilities. And we need to improve the use of technologies*”.

Moreover, Participant 3 suggested developing tools and databases at the health facility level to track medication dispensation, enabling more accurate forecasting. Leveraging real-time data on drug utilization was deemed vital for effective forecasting. “*Developing tools and database to see the actual use dispensed to patient is important and will solve most of the challenges related with the forecasting and EPSS should know every dispensed drug at the end user level and use that information for forecasting purpose*”.

Sub-theme 3: Workforce-related improvements 

The participants underscored the critical necessity of a proficient and accountable workforce within the healthcare supply chain to address overarching challenges, particularly in the realm of forecasting. Participant 8 eloquently articulated the need for a structured accountability framework akin to that observed in the banking sector, advocating for rigorous auditing and monitoring mechanisms to ensure transparency and responsibility among pharmacy professionals. They emphasized the pivotal role of capacity-building initiatives in equipping professionals with essential skills in forecasting, data management, and inventory control. “*We need to have skilled human resources who is accountable just like the bank officers are accounted for each money, and there is auditing each day in the bank with the same principle there should be auditing of the pharmaceuticals and they have to be accountable as the bankers are accountable for the money they will pay for the customers, the pharmacy professionals need to have computers and other important equipment’s. The professionals become reluctant because there is no auditing or check and balance*” (Participant 8).

Building upon this sentiment, Participant 15 emphasized the imperative of comprehensive capacity-building programs tailored to address skill deficiencies in RDF pharmaceutical management. They highlighted the pressing need to fortify data quality and inventory management practices at health facility levels, underscoring the pivotal role of training in fostering workforce competence and motivation. “*There should be capacity building training in RDF pharmaceuticals there are only few health facilities using bin cards so we need to improve data quality at health facilities level we need strengthen the inventory management at the health facilities*” (Participant 15). Moreover, participants unanimously emphasized the multifaceted benefits of training initiatives, not only in ameliorating skill gaps but also in bolstering workforce morale and engagement. Participant 6 underscored the transformative impact of training programs on enhancing both skill proficiency and intrinsic motivation among professionals, thereby contributing to sustained performance improvement within the supply chain domain. “*We need to provide training, which will help in solving the skill gap in forecasting but also it will increase their motivation*” (Participant 6).

While short-term training interventions were deemed crucial for immediate skill enhancement, participants emphasized the long-term imperative of elevating the quality of degree programs to cultivate a pipeline of skilled professionals. Participant 9 elucidated the dual-pronged approach of addressing immediate skill deficiencies through targeted training interventions while concurrently investing in the enhancement of academic curricula to ensure the sustained development of a competent workforce. “*In short term there should be training to improve the skill gap, in long run we need to improve the quality of the degree programme*” (Participant 9). Furthermore, the broader issue of workforce scarcity within the healthcare supply chain was acknowledged, with participants emphasizing the urgent need for comprehensive initiatives to augment both the quality and quantity of professionals in this domain. Participant 14 underscored the gravity of the challenge, highlighting the dearth of skilled professionals at the national level and advocating for concerted efforts to rectify this shortfall through strategic workforce development initiatives. “*In order to improve the problem in forecasting as well as in the health supply chain we need to work on human resources in terms of quality and quantity, the country does not have a skilled human resources in the field of forecasting at national level there are a few hand full of professionals and we need to act on these issue in order to solve the problem*” (Participant 14).

Sub-theme 4: Finance-related improvements

The participants underscored the pressing need for increased government financing in the pharmaceutical sector, recognizing the detrimental impact of financial shortages on both the supply chain and forecasting practices. This financial deficit was observed to significantly affect the procurement of pharmaceuticals, spanning both program medicines supported by external partners and the Revolving Drug Fund (RDF), where health facilities utilize a revolving budget model for pharmaceutical procurement. Central to the discourse was the imbalance between allocated financial resources and the requisite pharmaceutical quantities, prompting participants to propose various avenues for financial improvement.

Participant 10 advocated for proactive resource mobilization efforts to align forecasted pharmaceutical requirements with available financial resources, emphasizing the imperative of balancing supply projections with budgetary allocations. “*We need to mobilize the resources and we need to balance the forecasted quantity and the resources they have*”.

Similarly, Participant 2 highlighted the reliance on external funding sources and urged governments to preemptively address potential funding shortfalls to mitigate disruptions in pharmaceutical procurement processes. “*We always rely on external funds if these funds reduced or stop the governments should fill the gap. So the funding problem should address in well in advance*” (Participant 2).

Moreover, participants underscored the financial strain imposed by health facilities’ reliance on credit-based procurement from the Essential Pharmaceuticals Supply Services (EPSS), exacerbating financial burdens within the supply chain. Participant 10 proposed proactive measures to bolster the EPSS’s financial resilience by facilitating advance resource allocation, thereby alleviating financial pressures and streamlining procurement processes. “*The health facilities purchase the products from EPSS on credit based and these issues creates a huge burden on EPSS on the financial matters so if there is a possibility EPSS obtains their resource in advance it will help the procurement process to be easy. This will increase the financial capacity of EPSS*” (Participant 10). 

Sub-theme 5: Stakeholder-related improvements 

Stakeholder engagement emerged as a pivotal factor in enhancing forecasting activities within the healthcare supply chain, as highlighted by the participants. They emphasized the indispensable role of stakeholders not only in the broader context of healthcare supply chain management but also specifically in forecasting endeavors. The consensus among participants was that collaborative stakeholder efforts are imperative for optimizing forecasting practices and ultimately improving pharmaceutical supply chain efficiency.

Participants stressed the necessity for stakeholders to collaborate seamlessly and extend support to forecasting initiatives. Furthermore, they underscored the importance of integrating academic institutions into the collaborative framework to leverage theoretical expertise and augment operational effectiveness at both the EEPSS and Ministry of Health (MOH) levels. Participant 8 articulated this sentiment, advocating for comprehensive stakeholder engagement involving academia, healthcare facilities, and governmental entities: “*All stakeholders should come together, with higher institutions actively contributing to healthcare facilities. Academicians should be embedded within healthcare facilities, collaborating closely with MOH teams and logistics professionals across various stakeholders. Continuous learning and knowledge exchange among EPSS, MOH, partners, and academics are essential*”.

Participant 3 echoed this sentiment, emphasizing the need for stakeholder engagement in collaborative forecasting endeavors to harness expert insights and address forecasting challenges effectively. “*We need to engage stake holders and do the forecasting together these activities will improve the forecasting practice. This will help us to understand get expert opinions that will help the whole forecasting problem*” (Participant 3).

Additionally, participants highlighted the significance of stakeholder engagement in pharmaceutical supply chain management, emphasizing the importance of collaborative forecasting initiatives. Participant 4 underscored the potential benefits of involving stakeholders such as healthcare facilities and manufacturers in collaborative forecasting efforts to enhance supply chain management practices. “*To improve the Pharmaceutical supply chain management we need to engage the stakeholders like health facilities, manufacturers and other partners. It will be good if there is collaborative forecasting*” (Participant 4). 

## 5. Discussion

The Ethiopian pharmaceutical supply chain faces issues like lack of financing, workforce skills, poor coordination, data quality problems, and lack of end-to-end visibility. These factors create a systemic problem requiring an integrated solution. Bititci (2015) suggests that organizations perform better when working towards a common goal with collaboration and shared learning, enhancing problem-solving and innovation. This requires open communication, trust, and reflection [53]. This concept can be applied to the entire supply chain, aligning with responsible innovation theory, where stakeholders anticipate problems, act responsively, and learn collectively (Figure 2).

Embracing the principles of responsible innovation, characterized by transparent and collaborative processes, can guide effective decision-making [54]. Responsible innovation, as defined by Stilgoe et al., emphasizes the importance of considering the purpose and process of innovation alongside its outcomes. Four dimensions—anticipation, reflexivity, inclusion, and responsiveness—are integral to this framework [55]. Moreover, Naughton et al. suggest that “responsible impact” is the fifth domain and output of responsible innovation, potentially serving as a natural reinforcer of the responsible innovation process [56].

Anticipation involves governance mechanisms to foresee potential societal impacts [55,57]. There are many decisions that depend on the quality of forecasts in the healthcare system, from capacity planning to layout decisions to daily schedules. In general, the role of forecasting in healthcare is to inform both clinical and non-clinical decisions. While the former concern decisions related to patients and their treatments [58], the latter involve policy/management and supply chain decisions that support the delivery of high-quality care for patients. Forecasting is also used in both national and global healthcare supply chains, not only to ensure the availability of medical products for the population but also to avoid excessive inventory. Additionally, the lack of accurate demand forecast in a health supply chain may cost lives [59] and has exacerbated risks for suppliers [24].

Inclusivity aims to bring relevant stakeholders and the public to the table to engage with, identify, and address unforeseen, ungoverned, or ethical issues, as well as challenge assumptions and the innovation’s purpose, which is commonly the case with emerging technology and has been one of the most widely covered domains in recent years [60,61,62,63,64,65,66,67,68]. The issue of inclusivity is quite important in our findings, as we identified that coordination and collaboration were some of the challenges in pharmaceutical demand forecasting in the study area. Engaging relevant stakeholders is part of the problem, and working towards a common solution will improve most of the problems related to pharmaceutical forecasting.

Reflexivity is an important aspect of any innovation [57]. Stilgoe et al. describe it as holding a mirror up to oneself [55]. It is described as a process whereby one’s activities, considerations, commitments, and assumptions are challenged. This issue is quite important because we have witnessed a lack of commitment, poor data, and a lack of coordination and collaboration in the workforce while conducting forecasting activities, as evidenced in this study. We believe reflexivity cannot solve the problems of forecasting by itself; however, stakeholders need to collaborate and coordinate, which would require a lot of open information sharing, and reflexivity can enable the group to reflect on the actions they have taken, their outcomes, and on what worked and what did not. Thus, they will learn as a group [53]. Responsiveness involves reactions to influencing factors, which may include political and social agendas or the thoughts and concerns of stakeholders and the public [69]. It is acknowledged that the performance of healthcare operations results in the life or death of people, whereas it is only profit or loss in other sectors. In recent years, SC management (SCM) has become prevalent in the healthcare sector because of its impact on minimizing waste and medical errors, enhancing quality of care, and improving operational efficiencies and customer satisfaction [70]. In a complex system, you need to obtain as much real-time information as possible and take action as quickly as possible (responsiveness). If you do not, the system changes quickly and the planned action will no longer be valid; thus, the cycle of collaboration, coordination, sharing data, taking action, reflecting, and learning needs to be quite fast [53].

Responsible impact refers to the positive outcomes and effects that result from responsible innovation practices within an organization or ecosystem. The concept of responsible impact emphasizes the importance of considering the broader implications of innovation beyond economic gains. It highlights the need for organizations to assess and measure the positive outcomes of their innovation efforts in terms of sustainability, ethics, social responsibility, and overall impact on stakeholders and society. Recognizing and understanding responsible impact can help organizations align their innovation practices with ethical and sustainable principles, leading to more meaningful and beneficial outcomes for both the organization and the wider ecosystem in which they operate [56].

This study emphasizes the importance of responsibility and attention to forecasting to mitigate sector challenges. Gurzawska also highlights that responsible and sustainable SCM requires innovative technology, political solutions through multi-stakeholder partnerships, and ethical responsibility across supply chain tiers, necessitating a systemic approach [71]. To improve forecasting, responsible bodies should focus on technology, ensure end-to-end data visibility, and develop forecasting tools and databases. Digitalizing supply chains fosters communication and collaboration among stakeholders, but technological innovations alone are not enough without strong coordination [71]. Supply chains are multi-tier, involving manufacturers, intermediaries, end-users, governments, and communities. Effective SCM relies on stakeholder engagement and multi-stakeholder approaches (CSSPs) to achieve responsibility and sustainability [72,73,74,75]. This approach, rooted in stakeholder theory, asserts that a company’s success depends on sustainable relationships within its stakeholder network [76,77].

Freeman (1984) defines stakeholders as groups affecting or affected by an organization’s purpose [78]. The multi-stakeholder approach in SCM stems from the concept of collaborative enterprises, where companies build long-term, mutually beneficial relationships with stakeholders to create sustainable value [77]. This enhances relationships through better coordination with suppliers, customers, and other stakeholders, improving social outcomes and generating win-win solutions. Thus, a company’s sustainability depends on its stakeholder relationships [77,79]. This multi-stakeholder approach addresses collaboration, communication, and power distribution challenges in the supply chain. Collaborative initiatives enable better monitoring and tracing of supply chain activities, which individual companies may lack resources for the societal and environmental impacts of its activities [80]. However, developing these deep relationships requires time [81]. Non-state actors can enhance supply chain responsibility and sustainability by effectively gathering, collating, and providing information on policy issues and problem areas [82,83].

This paper highlights the interplay between technology, politics, and stakeholder involvement. Solutions like end-to-end data visibility, political will, and collaboration among supply chain actors could alleviate forecasting challenges. Non-state actor initiatives use technologies for data sharing via multi-stakeholder digital platforms, enhancing compliance through mutual control [71]. These platforms also promote transparency, cooperation, and trust, as well as reduce power abuse, requiring oversight for ethical decisions [84]. Effective governance involves multi-stakeholder collaboration for credibility, combining technological, political, and ethical solutions to address SCM challenges. This study highlights pharmaceutical forecasting challenges but has limitations. Its design may have introduced subjectivity and the small sample size limits generalizability. Data collection and interpretation biases also affect findings, making conclusions context-specific. Despite these constraints, this study is crucial for understanding and addressing forecasting challenges in resource-constrained settings.

## 6. Conclusions

In conclusion, this study highlights significant challenges within the Ethiopian public pharmaceutical supply chain, particularly in forecasting practices. These challenges, including finance-related constraints, workforce shortages, and data quality issues, present formidable barriers to efficient pharmaceutical procurement and distribution. Based on our discussions and the theories we covered in this paper, we believe the first priority needs to be resolving the problem of inaccuracy and unreliability in the supply chain. All stakeholders must work together as a single team to help address this problem. This will require certain incentives and disincentives to be introduced to motivate different stakeholders to collaborate and work as a team. A platform also needs to be created to enable the stakeholders to share information, communicate, and collaborate effectively in order to reinforce the cycle of thinking together, acting together, reflecting together and learning together. Finally, the learnings from this way of working need to be shared with the wider community, so that the valuable lessons and knowledge around what works well what does not work so well are not lost and continue to accumulate and improve the performance of the Ethiopian pharmaceutical supply chain. By integrating responsible innovation and impact principles into demand-forecasting practices, the EPSS can enhance the accuracy, reliability, and ethical considerations of their forecasting models. This approach can lead to more sustainable, patient-centered, and socially responsible decision-making in the pharmaceutical sector.

## Figures and Tables

**Figure 1 pharmacy-12-00086-f001:**
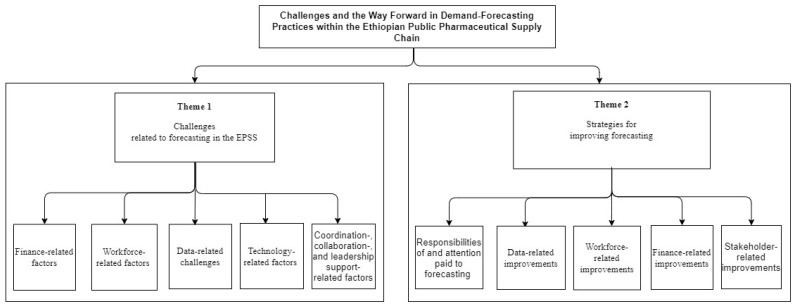
The main themes and sub-themes of the thematic analysis.

**Figure 2 pharmacy-12-00086-f002:**
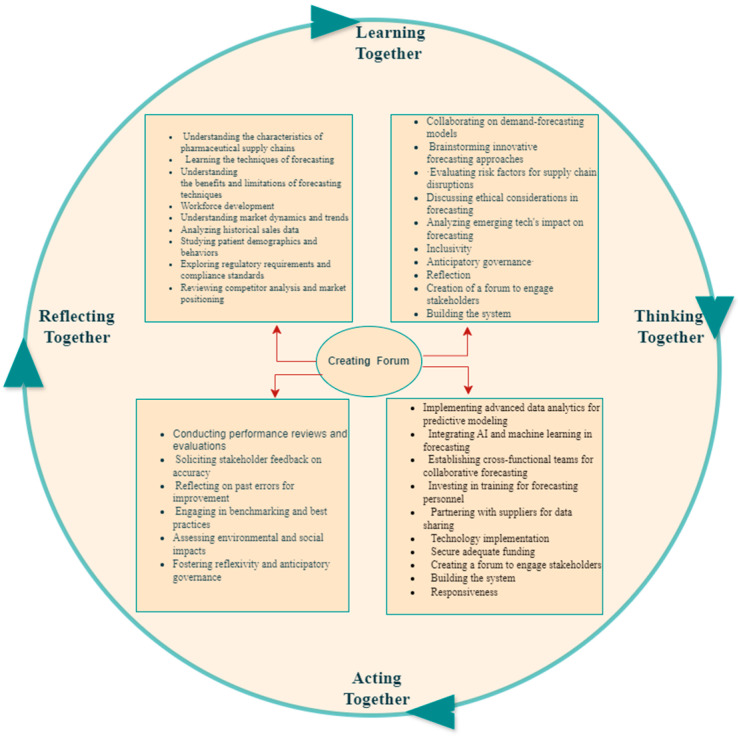
How EPSS can learn the forecasting process and mitigate its challenges. Sources: *Modification of Managing Business Performance: the Science and the Art*, Umit S. Bititci.

## Data Availability

Data can be obtained from the corresponding authors on request.

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
