# Peer review of "Challenges and the Way Forward in Demand-Forecasting Practices within the Ethiopian Public Pharmaceutical Supply Chain"

_pharmacy, 2024, doi:10.3390/pharmacy12030086_

Round 1
Reviewer 1 Report
Comments and Suggestions for Authors
Dear Authors,
Thank you for submitting this manuscript which I found to be an overall good paper and an interesting read. Overall, the paper is of good quality and will make a valuable contribution to the literature. However, there are a few points that are worth considering maximising the impact of this paper. Overall, the paper seems a bit on the long side and would benefit from being condensed.
Introduction:
· Generally speaking, the authors present a nice clear introduction which motivates the study.
· There is a sentence which states ‘. Even though there were 31 significant supports during the last decade, for addressing global health issues such as 32 HIV/AIDS, malaria, and TB in developing countries [9]’ This sentence appears to be incomplete. Please address this.
· On several occasions in the introduction, you use ‘etc.’ please avoid doing so and just list what you think is relevant. Do not assume the reader will know what else that list should contain.
Literature review:
· This is comprehensive and detailed but considering the length of the article I would suggest refining and condensing this section so as to maintain the reader’s attention.
Methods:
· Context: This section is very positive and provides some good detail but again I would err on the side of reducing this a little to make space elsewhere in the paper.
· Figure 1 is quite unclear so I would suggest you provide the original image to the journal to improve the image quality.
· Section 3.4 describes 11 participants and details who 9 of them were, who were the other 2? Then in the socio-demographic table you describe that there were 17 participants. Where did the additional 6 come from. Some further clarity regarding participant numbers and who they were, would be positive here.
· Section 3.5: Some of the questions you asked seem to be a little leading. For example, did you first broadly ask about pharmaceuticals access in Ethiopia, or did you as “What are the factors which affect the availability of pharmaceuticals in Ethiopia and in particular with EPSS?... The later question would be a bit leading and plant in the participants mind that there is a problem, whether or not they thought so previously. I would review the semi-structured interview questions and if some were a little leading those limitations should be added to the study limitations section.
· I didn’t notice you mention that you collected interviews until data saturation, was that the case, or was there a reason why you didn’t?
Results
· The table regarding socio-demographics of the participants is unlabelled and requires a label.
· Under the socio-demographic table is a description of the themes and sub-themes. A nice diagram to show the primary and sub themes would be great here as some readers prefer this and will ultimately illuminate your results and make them more memorable and easier to find in the paper for some readers.
· The results section is good but a bit long and could do with being condensed.
Discussion
· Compared to the results section the discussion is comparatively short. I would like to see some deeper engagement with your findings and to hear your thoughts regarding ways to improve the problem based on your results.
· Regarding Subtheme 5: Reading this I can’t help but think this aligns with the concept of responsible innovation. There are probably other theories/concepts you could lean on also, but responsible innovation would work here considering its emphasis on inclusivity, reflexivity, and anticipatory governance. By calling and referencing the value of stakeholder engagement, as responsible innovation your work could help not just pharmacy academics and practitioners, but also those interested in management and responsible innovation in healthcare. I suggest reading and including the early work by Stilgoe et al 2013 and position this stakeholder engagement piece as Responsible innovation in the discussion section. https://www.sciencedirect.com/science/article/pii/S0048733313000930 For examples of research on healthcare specific responsible innovation you could look at the work by Iakovleva et al. https://www.elgaronline.com/edcollchap/edcoll/9781788975056/9781788975056.00008.xml or Naughton et al. https://www.tandfonline.com/doi/full/10.1080/23299460.2023.2211870 This could add a little more weight to the discussion and provide the paper with a more concrete take home message. Also ask yourselves, what other approaches could be taken to help solve the problems identified in your paper?
· Also, regarding the discussion, your themes are well laid out but are there any links between those themes or are they totally unconnected? Is there a suitable framework from your themes you could propose? There doesn't have to be, but its worth asking the question now.
Conclusions
· The conclusions are fine but missing a bit if substance. I would like to see here a takeaway message which is interesting, surprising, or different which are of course backed up by the study data.
Author Response
Dear Reviewers,
We thank you for giving us the opportunity to revise our manuscript. We are also very much grateful for the constructive criticisms and suggestions forwarded by the reviewers, which in fact contributed a lot in improving the manuscript. We have attempted to incorporate as much as
Possible comments given by the referees (see below the point-by-point response) and hope the proposal is now in a form to be acceptable. .
Regards
Arebu Issa
Reviewer: # 1
Some specific comments:
Introduction
- There is a sentence which states ‘. Even though there were 31 significant supports during the last decade, for addressing global health issues such as 32 HIV/AIDS, malaria, and TB in developing countries [9]’ This sentence appears to be incomplete. Please address this.
- Thank you for your comments and it is edited accordingly
- Even though there was significant support during the last decade, for addressing global health issues such as HIV/AIDS, malaria, and TB in developing countries [9].
- On several occasions in the introduction, you use ‘etc.’ please avoid doing so and just list what you think is relevant. Do not assume the reader will know what else that list should contain
- Thank you and modified accordingly
Literature review:
- This is comprehensive and detailed but considering the length of the article I would suggest refining and condensing this section so as to maintain the reader’s attention.
- Thank you modified accordingly
Methods
- Context: This section is very positive and provides some good detail but again I would err on the side of reducing this a little to make space elsewhere in the paper.
- Thank you and modified accordingly
- Figure 1 is quite unclear so I would suggest you provide the original image to the journal to improve the image quality.
- As we edit the methods section and suggested by the reviewer to minimize the methods section so that we have agreed to remove the figure 1
- Section 3.4 describes 11 participants and details who 9 of them were, who were the other 2? Then in the socio-demographic table you describe that there were 17 participants. Where did the additional 6 come from. Some further clarity regarding participant numbers and who they were, would be positive here.
- Thank you and the explanation are given here
- A total of 17 participants were interviewed for this study in which 9 of them were from the regional health bureaus and 2 were from the ministry of health. The other participants where 4 of them were expert from EPSS and the rest 2 of the participants were from partner organizations. The total sum of the participants become 17
- Section 3.5: Some of the questions you asked seem to be a little leading. For example, did you first broadly ask about pharmaceuticals access in Ethiopia, or did you as “What are the factors which affect the availability of pharmaceuticals in Ethiopia and in particular with EPSS?... The later question would be a bit leading and plant in the participants mind that there is a problem, whether or not they thought so previously. I would review the semi-structured interview questions and if some were a little leading those limitations should be added to the study limitations section.
- Thank you for the question we approached our participants by asking their socio demographic characteristics followed by general questions which addressed the supply chain for example How do you see the health supply chain in Ethiopia and in particular in EPSS (Probe on what are the problems). Then we furtherer ask the respondents deep questions which address factors which affect the availability of pharmaceuticals in Ethiopia. As you raised it the way we write in the paper looks like it is a leading question in order to prevent that we have included some of the points and rearranged accordingly.
- Moreover we have include limitation of the study in the manuscript
- I didn’t notice you mention that you collected interviews until data saturation, was that the case, or was there a reason why you didn’t?
- Thank you Yes indeed we collected the data collection until saturation points reached and we have included the points it in the revised section in the data collection methods and procedures
Results
- The table regarding socio-demographics of the participants is unlabelled and requires a label.
- Modified accordingly
- Under the socio-demographic table is a description of the themes and sub-themes. A nice diagram to show the primary and sub themes would be great here as some readers prefer this and will ultimately illuminate your results and make them more memorable and easier to find in the paper for some readers.
- Thank you and modified accordingly
- The results section is good but a bit long and could do with being condensed.
- Thank you so much we have modified it accordingly and tried to shorten the result section
Discussion
- Compared to the results section the discussion is comparatively short. I would like to see some deeper engagement with your findings and to hear your thoughts regarding ways to improve the problem based on your results.
- Thank you for the good insight and we have taken the suggestions and modified the discussions section accordingly and please the discussions sections in the manuscript
- Regarding Subtheme 5: Reading this I can’t help but think this aligns with the concept of responsible innovation. There are probably other theories/concepts you could lean on also, but responsible innovation would work here considering its emphasis on inclusivity, reflexivity, and anticipatory governance. By calling and referencing the value of stakeholder engagement, as responsible innovation your work could help not just pharmacy academics and practitioners, but also those interested in management and responsible innovation in healthcare. I suggest reading and including the early work by Stilgoe et al 2013 and position this stakeholder engagement piece as Responsible innovation in the discussion section. https://www.sciencedirect.com/science/article/pii/S0048733313000930 For examples of research on healthcare specific responsible innovation you could look at the work by Iakovleva et al. https://www.elgaronline.com/edcollchap/edcoll/9781788975056/9781788975056.00008.xml or Naughton et al. https://www.tandfonline.com/doi/full/10.1080/23299460.2023.2211870 This could add a little more weight to the discussion and provide the paper with a more concrete take home message. Also ask yourselves, what other approaches could be taken to help solve the problems identified in your paper?
- Thank you for suggesting references; we have incorporated them into our theoretical framework. The theory of responsible innovation played a significant role in shaping our discussions and findings
- .Also, regarding the discussion, your themes are well laid out but are there any links between those themes or are they totally unconnected? Is there a suitable framework from your themes you could propose? There doesn't have to be, but its worth asking the question now.
- Thank you and we have modified according to your suggestions and indicated in the discussions sections of the manuscript
Conclusions
- The conclusions are fine but missing a bit if substance. I would like to see here a takeaway message which is interesting, surprising, or different which are of course backed up by the study data.
- Thank you we have modified the conclusions sections accordingly

Reviewer 2 Report
Comments and Suggestions for Authors
The authors comprehensively introduced the challenges in demand forecasting of pharmaceutical supply chain. Moreover this analysis well designed the study to collect the inputs of professionals to resolve the problems of pharmaceutical forecasting. However, the interview questions may be limited and misleading. A more comprehensive multi-choice questionnaire may be contributive.
Please review the logical flow of ideas and transitions between paragraphs in section 2 and 3. The coherence and cohesion in the section can be improved to ensure a smooth reading experience.
Figure 1 is hard to read, please modify the figure for appropriate size and resolution.
Some summarized tables of themes/questions/answers and thematic analysis maps can be provided to improve the explanation and discussion of the results.
Comments on the Quality of English LanguageThere are lots of grammatical errors and punctuation mistakes in the paper.
The sentence structure and overall coherence in section 2 need to be improved. For example, the challenges in section 2.1 are not enumerated properly from line 84.
Please eliminate redundant phrases in section 3 to tighten the text
Author Response
Dear Reviewers,
We thank you for giving us the opportunity to revise our manuscript. We are also very much
grateful for the constructive criticisms and suggestions forwarded by the reviewers, which in fact contributed a lot in improving the manuscript. We have attempted to incorporate as much as possible comments given by the referees (see below the point-by-point response) and hope the proposal is now in a form to be acceptable. .
Regards
Arebu Issa
Reviewer: # 1
Some specific comments:
Comments and Suggestions for Authors
- The authors comprehensively introduced the challenges in demand forecasting of pharmaceutical supply chain. Moreover this analysis well designed the study to collect the inputs of professionals to resolve the problems of pharmaceutical forecasting. However, the interview questions may be limited and misleading. A more comprehensive multi-choice questionnaire may be contributive.
- Thank you for your comments we used a key informant interview because it will help us to understand the challenges in pharmaceutical demand forecasting’s and we will consider for the next time to make it a multiple choice one.
- Please review the logical flow of ideas and transitions between paragraphs in section 2 and 3. The coherence and cohesion in the section can be improved to ensure a smooth reading experience.
- Thank you for your comments we have modified the section to make it more coherent
Figure 1 is hard to read, please modify the figure for appropriate size and resolution.
- Thank you for the comments we have removed figure 1 when we are minimizing the methods section which is commented by the other reviewer as the section is a little bit wider.
- Some summarized tables of themes/questions/answers and thematic analysis maps can be provided to improve the explanation and discussion of the results.
- Thank you for your comments we had modified it accordingly by inserting a a diagrams to summarize the themes and sub themes in the study
Comments on the Quality of English Language
- There are lots of grammatical errors and punctuation mistakes in the paper.
- Thank you the manuscript is edited by one of the author who is a native English speaker and modified accordingly
The sentence structure and overall coherence in section 2 need to be improved. For example, the challenges in section 2.1 are not enumerated properly from line 84.
Please eliminate redundant phrases in section 3 to tighten the text
- Thank you and modified accordingly

Round 2
Reviewer 1 Report
Comments and Suggestions for Authors
Dear Authors,
Thank you for your re-submission. I am happy with the paper. However, there are a few small remaining issues that require resolution.
- Line 729 you classify the four dimensions of responsible innovation. Research research has updated this idea and state that there are five dimensions and the fifth is 'Responsible impact' as outlined in the following paper.
Naughton, B., Dopson, S., & Iakovleva, T. (2023). Responsible impact and the reinforcement of responsible innovation in the public sector ecosystem: cases of digital health innovation. Journal of Responsible Innovation, 10(1). https://doi.org/10.1080/23299460.2023.2211870
To ensure your arguments are up to date with the latest literature I would include and reference this 'Responsible Impact' dimension as it will add value to the paper, as well as demonstrating that your paper is up to date with the most recent research in this area.
- Also do you have copyright permission to share figure 2? If you don't have this you will have to re-produce your own version of that diagram. Also it is very pixilated and unclear so a different version will be required in any case.
- As you have included responsible innovation in the discussion it should feature, in some small way, in the conclusions.
- Finally, please try to cut the length overall as it seems a bit on a the long side.
All the very best,
Author Response
Dear Reviewers,
Thank you for the opportunity to revise our manuscript. We greatly appreciate the constructive criticisms and suggestions provided by the reviewers, which have significantly improved the manuscript. We have incorporated the comments from the referees as thoroughly as possible (see our point-by-point response below) and hope the revised proposal is now acceptable.
Regards,
Arebu Issa
Reviewer: # 1
Some specific comments:
- Thank you for your re-submission. I am happy with the paper. However, there are a few small remaining issues that require resolution.
- Line 729 you classify the four dimensions of responsible innovation. Research research has updated this idea and state that there are five dimensions and the fifth is 'Responsible impact' as outlined in the following paper.
Naughton, B., Dopson, S., & Iakovleva, T. (2023). Responsible impact and the reinforcement of responsible innovation in the public sector ecosystem: cases of digital health innovation. Journal of Responsible Innovation, 10(1). https://doi.org/10.1080/23299460.2023.2211870
To ensure your arguments are up to date with the latest literature I would include and reference this 'Responsible Impact' dimension as it will add value to the paper, as well as demonstrating that your paper is up to date with the most recent research in this area.
- Thank you for your suggestion and insights. We have added the references and incorporated the ideas from the paper.
- Also do you have copyright permission to share figure 2? If you don't have this you will have to re-produce your own version of that diagram. Also it is very pixilated and unclear so a different version will be required in any case.
- One of the co-authors holds the copyright for the diagram, which has been modified and is not identical to the original. We have now produced an improved version of the diagram and included it in the document.
- - As you have included responsible innovation in the discussion it should feature, in some small way, in the conclusions.
- Thank you for the suggestions. We have included them in the conclusions section.
- Finally, please try to cut the length overall as it seems a bit on the long side.
- Thank you. We have made significant efforts to reduce the word count from approximately 12,000 to 10,700 words.
